# BCG epidemiology supports its protection against COVID-19? A word of caution

Reka Szigeti[1], Domos Kellermayer[2], Giedrius Trakimas[3,4], Richard Kellermayer [5,6,7] *

**1** Department of Pathology & Genomic Medicine, Houston Methodist Hospital, Houston, Texas, United States of America, **2** University of Houston, Houston, Texas, United States of America, **3** Institute of Biosciences, Vilnius University, Vilnius, Latvia, **4** Institute of Life Sciences and Technology, Daugavpils University, Daugavpils, Latvia, **5** Section of Pediatric Gastroenterology, Texas Children's Hospital, Houston, Texas, United States of America, **6** Baylor College of Medicine, Houston, Texas, United States of America, **7** USDA/ARS Children's Nutrition Research Center, Houston, Texas, United States of America

* kellerma@bcm.edu

**Data Availability Statement:** All relevant data are within the manuscript and its Supporting Information files.

**Funding:** The authors received no specific funding for this work.

## Abstract

The COVID-19 pandemic, caused by type 2 Severe Acute Respiratory Syndrome Coronavirus (SARS-CoV-2), puts all of us to the test. Epidemiologic observations could critically aid the development of protective measures to combat this devastating viral outbreak. Recent observations, linked nation based universal Bacillus Calmette-Guerin (BCG) vaccination to potential protection against morbidity and mortality from SARS-CoV-2, and received much attention in public media. We wished to validate the findings by examining the country based association between COVID-19 mortality per million population, or daily rates of COVID-19 case fatality (i.e. Death Per Case/Days of the endemic [dpc/d]) and the presence of universal BCG vaccination before 1980, or the year of the establishment of universal BCG vaccination. These associations were examined in multiple regression modeling based on publicly available databases on both April 3rd and May 15th of 2020. COVID-19 deaths per million negatively associated with universal BCG vaccination in a country before 1980 based on May 15th data, but this was not true for COVID-19 dpc/d on either of days of inquiry. We also demonstrate possible arbitrary selection bias in such analyses. Consequently, caution should be exercised amidst the publication surge on COVID-19, due to political/economical-, arbitrary selection-, and fear/anxiety related biases, which may obscure scientific rigor. We argue that global COVID-19 epidemiologic data is unreliable and therefore should be critically scrutinized before using it as a nidus for subsequent hypothesis driven scientific discovery.

## Introduction

There is a current global crisis from the Coronavirus Disease of 2019 (COVID-19) pandemic [1]. COVID-19 is caused by type 2 Severe Acute Respiratory Syndrome Coronavirus (SARS-CoV-2), which is a medium-sized, enveloped, positive-stranded RNA virus of the *Coronaviridae* family. SARS-CoV-2 is the pathogen of the third, large, severe respiratory syndrome outbreak caused by Coronaviruses (CoVs) (1: SARS [severe acute respiratory syndrome] which emerged in late 2002 and disappeared by 2004; 2: MERS [Middle East respiratory syndrome], which emerged in 2012 and remains in circulation in camels) [2].

**Competing interests:** The authors have declared that no competing interests exist.

COVID-19 cases and associated deaths continue to rise [3], which naturally induces fear, anxiety and sadness in all of us. Time is of essence towards finding definite solutions for stopping the pandemic, and scientists are under significant pressure trying to balance speed with safety and precision [4]. Since fear and sadness can alter our cognitive control [5], there is valid concern about loosened scientific rigor in respect to the massive surge of publications amidst the time pressure on biomedical scientists racing for a cure. Even though the outbreak likely began in December of 2019 in Wuhan of Hubei Province in China [6], there are ongoing uncertainties about SARS-CoV-2 epidemiology. Rigorous studies (including multiple site and repeated nucleic acid based-, and also viral culture based testing) in 9 symptomatic patients with mild disease course have shown active viral replication in the upper airway, and high viral shedding in pharynx (but lower than in sputum) peaking at 4 days of symptoms [7]. SARS-CoV-2 virus was readily isolated from throat- and lung-derived samples, but not from stool, in spite of high virus RNA concentration in the fecal samples [7]. Blood and urine never yielded live virus [7]. On the contrary, among the first cases in Europe, at least RNA based viremia was detected in a severe form of the disease progressing to multi-organ failure [8]. Investigators from China found the highest SARS-CoV-2 viral load in throat swabs at the time of symptom onset, and inferred that infectiousness peaked on or before symptom onset [9]. These findings underscore the primary respiratory spread of SARS-CoV-2, at least during mild disease course, which has been traditionally considered to be large droplet/contact communicated based on findings with SARS and MERS. However, some work indicates the potential airborne spread of the virus [10], leading to debates amongst infectious disease experts between contact vs. airborne protection recommendations. Adding to the difficulties in making clear cut regulations for personal protective equipment (PPE) use is viral shedding before symptomatic presentation [9]. Importantly, a large proportion of infected people can be asymptomatic who can spread the virus [11]. Such viral spreading may be prominent in children and young adults, with over 50% being asymptomatic or having mild disease [12, 13]. Additionally, some patients present with gastrointestinal complaints, and never develop respiratory symptoms [14], for example. Furthermore, due to limitations in nucleic acid based analysis including quality of sample collection and variable methodology [15], a single nasopharyngeal swab can have as low as 32% sensitivity over the course of infection [16]. In the meantime, an asymptomatic patient may have similar viral loads as symptomatic ones, indicating the transmission potential from such cases [17]. These observations add to the tremendous difficulties in developing clear and consistent guidelines for PPE use during the COVID-19 pandemic, especially in the medical setting.

As already mentioned, similarly to MERS [18] and SARS [19], pediatric patients with COVID-19 run a much milder disease course than the elderly (especially above 60 years of age) [20]. The exact reason for this is unknown [13], but at least in non-human primates with experimental SARS-CoV infection, immune responses (most prominently CD8 T cell and B cell associated) were greatly reduced in the aged host compared to younger animals [21]. Consequently, a number of immunomodulatory treatments are being explored to help patients in fighting the infection. Amidst these explorations, the effect of prophylactic Bacillus Calmette-Guerin (BCG) vaccination on COVID-19 outcomes was being investigated in 14 registered clinical trials (on clinicaltrials.gov) as of June 16[th] 2020. Linking to these investigations is a not yet peer-reviewed paper [22], suggesting a connection between universal BCG vaccination policy and the peculiarly significant nation based variation in case frequency and death rates from COVID-19, based on data from March 21st, 2020. This manuscript received both public and scientific attention, supporting [23] and contradicting [24] it. A more recently published work found significant correlation between an arbitrarily designed, country specific BCG index and COVID-19 mortality, following attempts to control for confounding variables [25]. The

epidemiology of the pandemic, however, is in ongoing flux. Therefore, we decided to examine this question with a modified definition of COVID-19 death rate based on both April 3[rd], and May 15[th] 2020 data.

## Methods

COVID-19 epidemiologic data was extracted from [3] on the afternoon of April 3[rd], 2020 for the top 68 countries based on number of cases. Epidemiologic data was repeatedly extracted from [3] on May 15[th] of 2020. Further demographic data (i.e. median age, population density, percentage of urban population) was extracted from https://www.worlometers.info/population/ [26] on May 15th, 2020. Data on air passengers (a surrogate value of mobility)—registered carrier departures worldwide was extracted from https://data.worldbank.org/indicator/IS.AIR.DPRT. Day of country dependent "onset" was defined as first confirmed case reported and extracted from https://ourworldindata.org/coronavirus. Total number of days from day of onset to April 3[rd], and May 15[th] 2020 was calculated for each country. Due to tremendous variation in population based testing (36/million in Indonesia to 74,416/million in Iceland on April 3[rd] 2020) and the importance of time between diagnosis and death, we arbitrarily defined death rate as Death Per Case (i.e. case fatality)/Days (dpc/d) for the endemic of each country.

Data on BCG vaccination was extracted from the BCG World Atlas [27] similarly as in [22], or from online searches for those few countries, which were not analyzed in the Atlas, one example being Iceland [28]. Modern "Colonial Era" countries to colonize America and Africa were defined as: Netherlands, Spain, United Kingdom, France, Belgium, Portugal, and Germany. We examined these countries as ones with 'historic colonization status' to highlight the significant potential for selection bias in the COVID-19 global epidemiologic data.

As opposed to Miller, et al. [22], we did not exclude countries with a population less than 1 million from our analyses, arguing that smaller countries may actually have better policies for universal testing (i.e. supporting rigorous epidemiologic analyses) than larger ones, Iceland being the prime example. Rather, we decided to study the top 68 countries for number of cases reported on April 3[rd]. This subjective cut-off was made for the Diamond Princess Cruise ship included in the list ranking at 68th (the ship's data was excluded). Amongst these 68 countries, we could identify the initiation year of universal BCG vaccination in 40 (S1 File of S1 Table). Out of the countries examined, 9 did not have universal BCG vaccination before 1980 (S1 File of S1 Table), which date we arbitrarily selected as the cutoff for having BCG vaccination "introduced" in respect to COVID-19 (since that would have affected the population of a country 40 years old and above [i.e. the population with increased vulnerability towards the infection]).

As for the May 15[th], 2020 dataset, we arbitrarily examined those countries which had more than 1,000 cases reported by that time.

We fitted two multiple regression models with the April 3, 2020 data-set using mortality rates (death/million OR dpc/days) as dependent variables for each model respectively, and BCG vaccination status before 1980, historic colonization status, median age, urban population percentage, population density, and air passengers as independent variables (predictors or confounders, depending on how one approaches the question). We repeated the analyses using the May 15, 2020 dataset including tests/million as additional independent variable (S1 File of S2 Table). The death/million and dpc/d variables were square-root transformed (SQRT), while population density, air passengers, and tests/million were log-transformed in order to achieve normality and decrease heteroscedasticity [29]. In order to compare the predictors' relative importance on the mortality rates, we reported standardized regression coefficients. Spearman's rank-order correlations were used to assess the relationships between the

year of the establishment of universal BCG vaccination and mortality rates (death/million or dpc/days) using April 3 and May 15 data sets. All statistical tests used in this study were two-tailed. Results were considered significant if p < 0.05. Analyses were performed using IBM SPSS 22 for Windows.

## Results

There were no significant correlations between the year of the establishment of universal BCG vaccination and mortality rates: death/million or dpc/days using April 3, 2020 data (rs = -.216, p = .18, n = 40 and rs = -.052, p = .751, n = 40, respectively). There was a significant negative correlation between the year of the establishment of universal BCG vaccination and death/million for the May 15 data-set (rs = -.28, p = .035, n = 57). However, there was no significant correlation between the year of the establishment of universal BCG vaccination and the dpc/days (rs = -.20, p = .135, n = 57) for the same, more recent data set.

Following direct correlation analyses, we proceeded with multiple regression analyses. We first examined deaths/million as the dependent variable for COVID-19 mortality. For the April 3 data-set, multiple regression statistically significantly predicted deaths/million, F(6, 61) = 10.181, p < .001, $R^2$ = .50, Adj. $R^2$ = .451 (Table 1). Arbitrary 'historic colonization status', BCG vaccination status before 1980, and median age added statistically significantly to the prediction model (p < .001, p = .004 and p = .015, respectively), while urban percentage, population density, and air passengers were non-significant, p > .05 (Table 1).

For the May 15 data-set multiple regression model, F(7, 84) = 15.726, p < .001, $R^2$ = .58, Adj. $R^2$ = .545 (Table 1), significant predictors of COVID-19 deaths/million were historic colonization status (p < .001), BCG vaccination status before 1980 (p = .002) and median age (p = .045), while, urban percentage, population density, tests/million and air passengers were non-significant, p > .05 (Table 1).

Table 1. Parameter estimates for predictors of mortality (SQRT death/million) in models of April 3 (n = 68) and May 15 (n = 92).

| Predictors | Standardized coefficients | t | p | Model |
| --- | --- | --- | --- | --- |
| | | | | Adj. $R^2$ |
| **April 3** | | | | .451 |
| Historic colonization status | .481*** | 4.795 | < .001 | |
| BCG vaccination before 1980 | -.280** | -3.012 | .004 | |
| Median age | .243* | 2.498 | .015 | |
| Population density (LOG) | .024 | .253 | .801 | |
| Air passengers (LOG) | -.023 | -.249 | .804 | |
| Urban percentage | .020 | .212 | .833 | |
| **May 15** | | | | .545 |
| Historic colonization status | .502*** | 6.497 | < .001 | |
| BCG vaccination before 1980 | -.227** | -3.151 | .002 | |
| Median age | .201* | 2.039 | .045 | |
| Tests/million (LOG) | .170 | 1.662 | .100 | |
| Air passengers (LOG) | .097 | 1.252 | .214 | |
| Population density (LOG) | -.072 | -.993 | .323 | |
| Urban percentage | -.019 | -.220 | .826 | |

SQRT indicates square-root transformation, LOG indicates $\log_{10}$ transformation. Asterisks mark significant coefficients:

*p < 0.05

**p < 0.01

***p < 0.001.

**Table 2. Parameter estimates for predictors of mortality (SQRT dpc/days) in models of April 3 (n = 68) and May 15 (n = 92).**

| Predictors | Standardized coefficients | t | p | Model Adj. $R^2$ |
|---|---|---|---|---|
| **April 3** | | | | .139 |
| Historic colonization status | .374** | 2.978 | .004 | |
| Urban percentage | -.230 | -1.925 | .059 | |
| Median age | -.210 | -1.723 | .090 | |
| BCG vaccination before 1980 | -.193 | -1.655 | .103 | |
| Air passengers (LOG) | -.079 | -.668 | .507 | |
| Population density (LOG) | -.037 | -.317 | .752 | |
| **May 15** | | | | .387 |
| Median age | .620*** | 5.426 | < .001 | |
| Tests/million (LOG) | -.596*** | -5.023 | < .001 | |
| Historic colonization status | .348*** | 3.872 | < .001 | |
| Population density (LOG) | -.155 | -1.833 | .07 | |
| BCG vaccination before 1980 | -.132 | -1.579 | .118 | |
| Urban percentage | -.057 | -.562 | .575 | |
| Air passengers (LOG) | .043 | .482 | .631 | |

SQRT indicates square-root transformation, LOG indicates $\log_{10}$ transformation, dpc indicates death per case (i.e. case mortality), days indicates days between first case reported and the date the analysis was performed (i.e. the reported length of the epidemic by country). Asterisks mark significant coefficients:

**$p < 0.01$

***$p < 0.001$.

We then examined dpc/d as the dependent variable for COVID-19 mortality by multiple regression analysis. The April 3 data model explained a relatively small amount of dpc/days variation F(6, 61) = 2.805, p = .018, $R^2$ = .216, Adj. $R^2$ = .139, with the historic colonization status being the only significant predictor (p = .004) (Table 2). For the May 15 data-set, however, the multiple regression model explained more variation: F(7, 84) = 9.206, p < .001, $R^2$ = .434, Adj. $R^2$ = .387. Significant predictors of May 15 dpc/days were test/million, median age, and historic colonization status (all p < .001), while BCG vaccination status before 1980, urban percentage, population density, and air passengers were non-significant, p > .05 (Table 2).

## Discussion

In this study, we found no significant association between universal BCG vaccination and country based COVID-19 mortality variation as defined by an arguably more precise death rate (i.e. dpc/d) definition than simply death/million, as examined by Miller, et al. [22], or by Escobar, et al. [25]. We underscore that both testing for-, and reporting of death from COVID-19 is highly influenced by nation specific political, cultural, and socioeconomic bias. Such bias is exemplified by no reported cases in North Korea, underreported cases and deaths in Yemen [30], very likely underreporting of deaths in various countries [31], and exploiting the pandemic for political gains [32], just to name a few. Consequently, the worldwide epidemiologic data is highly unreliable, which specifically pertains to COVID-19 deaths per million death rate, by our opinion. Such simple death rates are bound to have the most prominent political influence, and therefore are most vulnerable to bias. In the meantime, once a COVID-19 positive case is officially reported, a dependent obligation is created to provide outcomes for the case, less influenced by political, social, or cultural bias. Such bias is difficult to enumerate or account for, and current "evidence" for it in respect to COVID-19 relies on

social media. Therefore, it is 'unscientific'. Pretending for scientific scrutiny, however, by using country specific death rate or mortality as an "objective" COVID-19 outcome is misleading. In the meantime, such highly biased epidemiology based conclusions may be used as "evidence" for hypotheses, such as infantile/pediatric BCG vaccination providing lifelong protection against COVID-19 complications, in this case. For these reasons, we underscore that case mortality by length of the endemic for each country, or death per case per day (i.e. our dpc/d) measure is a more reliable measure of COVID-19 case severity than death rate alone. This conclusion is supported by our observation that tests/million (i.e. a rather objective parameter) was a significant predictor of dpc/d based COVID-19 severity for the more recent May 15th data set, but not of death/million severity outcome.

BCG vaccination is currently performed shortly after birth (newborns) in most of the countries, which universally vaccinate, as a protective measure against infantile/pediatric tuberculosis. There are some animal model and observational studies indicating that BCG vaccination can modulate host immunity (designated as 'trained immunity') and may protect against non-mycobacterial respiratory (and even other) infections as well (off target effect), especially in early childhood (reviewed in [33]). However, the effects of BCG vaccination fades with age, even against tuberculosis [34]. Hence, many countries have stopped their universal newborn vaccination programs or never even started that, since infantile/pediatric tuberculosis has become very rare in the economically advanced world. Consequently, there is no biologic evidence that newborn/baby age delivered BCG vaccination may have any protective effects against COVID-19, especially in adults and the elderly (who have received the BCG vaccine as an infant or child). Importantly, Hamiel, et al. did not find any difference in COVID-19 infection rates or case severity between similarly aged young adults who were either BGG immunized or not as infants in Israel [35].

The question whether BCG vaccination may acutely protect against COVID-19 infection and/or complications is very different form the one this paper addresses. We simply claim that this question cannot be answered through epidemiologic observations on historical (pediatric age delivered), country based BCG vaccination policy and associated COVID-19 endemic outcomes. Nevertheless, the most convincing support for the "acute BCG protection hypothesis" comes from a very recently published, double blind placebo controlled trial. Giamarellos-Bourboulis, et al. [36] enrolled recently hospitalized elderly (>65y old) patients to receive BCG vaccination (strain 1331; Intervax), or placebo (0.1.ml normal saline) by intradermal injection in a double blinded, randomized fashion. The primary outcome was the time interval to the first infection post hospital. Patients were followed for 12 months. Most importantly, significant protection against respiratory tract infections of probable viral origin (hazard ratio 0.21, p: 0.013) was observed in the BCG group. The investigators, however, did not clearly describe how blinding of patients and study personnel could be achieved in this case. Most commonly, a skin reaction occurs in 10–14 days at the site of the BCG injection and a permanent small scar develops in 3–6 months after the immunization in more than 85% of adults receiving the vaccine [37]. Therefore, it is biologically not possible to blind BCG with normal saline placebo. This oversight in one of the highest impact scientific journals (i.e. Cell) repeatedly emphasizes the significant bias that surrounds the COVID-19 pandemic at all levels of biomedicine from research to publication.

It is important to recognize that universal BCG vaccination establishment and current policy is a highly dependent variable, commonly inversely correlating with country specific economic status. Consequently, it is the economically most advanced countries, frequently with the most established democracies, which have abandoned or never established (such as the USA) universal BCG vaccination. We speculate that these countries are actually the ones where SARS-CoV-2 testing is most widespread and the reporting of cases and deaths is the

most transparent. Historically, these countries were commonly those, which participated in colonizing other parts of the world since the late 1400s, designated as Modern Era Colonizers ('historic colonization status' predictor in the Tables). Therefore, to demonstrate the arbitrary selection bias (i.e. post hoc explanation for variable COVID-19 severity) in universal (pediatric age) BCG vaccination modulating COVID-19 death rates decades after its delivery, we included historic colonization status as an "independent" variable/predictor into our multiple regression models for COVID-19 outcomes (death/million vs. dpc/d). This arbitrary 'historic colonization status' variable actually turned out to be a much stronger predictor for both of the COVID-19 mortality measures than BCG vaccination status.

Speculative biologic explanation (similar to that to newborn BCG vaccination) for Colonizer countries having higher mortality rates from COVID-19 could be generated (long standing, transgenerational influence of improved prenatal nutrition [38] on postnatal immune responses and life expectancy [leading to increased vulnerability to COVID-19] in these richer countries compared to the rest of the world, for example). In the meantime, we rather conclude that both BCG vaccination and Colonizer status are dependent variables of country based socioeconomic and political status, which latter features are the strongest predictors for COVID-19 outcomes (especially death/million) due the highly politicized nature of the pandemic.

Our work highlights the difficulties in drawing reliable epidemiologic conclusions from the currently available worldwide data on the COVID-19 pandemic. We advise for extreme caution and self-reflective scrutiny to balance publication pressure, inherent drive for scientific discovery, and financial social and political gains, when examining COVID-19 related biomedical research, including epidemiology. This conclusion is in line with experts in the field, emphasizing that "the data is not from peer-reviewed research, but rather is almost real-time clinical data–which can be messy and come with many caveats" [39]. The experts also underscore that "the lack of widespread, systematic testing in most countries is the main source of discrepancies in death rates internationally" [39]. It is our responsibility, as the medical scientific community around the world, to promote consistent and reliable epidemiologic reporting, and de-politicizing the current COVID-19 pandemic in order to prepare for other expectable large-scale infectious outbreaks in the future.

## Supporting information

**S1 File.**
(XLSX)

## Author Contributions

**Conceptualization:** Richard Kellermayer.

**Data curation:** Reka Szigeti, Domos Kellermayer, Giedrius Trakimas, Richard Kellermayer.

**Formal analysis:** Domos Kellermayer, Giedrius Trakimas, Richard Kellermayer.

**Funding acquisition:** Reka Szigeti.

**Supervision:** Richard Kellermayer.

**Writing – original draft:** Reka Szigeti, Richard Kellermayer.

**Writing – review & editing:** Reka Szigeti, Giedrius Trakimas, Richard Kellermayer.

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
