## [Decision Letter · Decision Letter 0]

4 May 2020

PONE-D-20-10246

BCG protects against COVID-19? A word of caution

PLOS ONE

Dear Dr. Kellermayer,

Thank you for submitting your manuscript to PLOS ONE. After careful consideration, we feel that it has merit but does not fully meet PLOS ONE’s publication criteria as it currently stands. Therefore, we invite you to submit a revised version of the manuscript that addresses the points raised during the review process.

Specifically, please take in account the limitation and advance highlighted by the two reviewers.

We would appreciate receiving your revised manuscript by Jun 18 2020 11:59PM. To enhance the reproducibility of your results, we recommend that if applicable you deposit your laboratory protocols in protocols.io, where a protocol can be assigned its own identifier (DOI) such that it can be cited independently in the future. For instructions see: http://journals.plos.org/plosone/s/submission-guidelines#loc-laboratory-protocols

We look forward to receiving your revised manuscript.

Kind regards,

Pierre Roques, Ph.D.

Academic Editor

PLOS ONE

Journal Requirements:

'NO - Include this sentence at the end of your statement: The funders had no role in study design, data collection and analysis, decision to publish, or preparation of the manuscript'

Reviewers' comments:

Reviewer's Responses to Questions

**Comments to the Author**

1. Is the manuscript technically sound, and do the data support the conclusions?

Reviewer #1: Partly

Reviewer #2: Yes

2. Has the statistical analysis been performed appropriately and rigorously? 

Reviewer #1: No

Reviewer #2: Yes

3. Have the authors made all data underlying the findings in their manuscript fully available?

Reviewer #1: No

Reviewer #2: Yes

4. Is the manuscript presented in an intelligible fashion and written in standard English?

Reviewer #1: Yes

Reviewer #2: Yes

5. Review Comments to the Author

Reviewer #1: The article from Richard Kellermayer’s group presents a critical analysis of the recent publication by Miller et al 2020 ref. [15] and proposes an original approach to establish a representation of the daily rates of COVID-19 case fatality (i.e. Death Per Case /Days of the endemic [dpc/d]). However, only strictly identical groups (age/sex proportion, living conditions, COVID diagnosis) could be strictly compared and would better answer the question of the effect of the BCG vaccine on current acute epidemic infection; could you address this question by adding an analysis of equivalent groups (for ex. in urban conditions)? Furthermore, to support the article statements, it is necessary to discuss whether the people who have benefited from BCG vaccination or not are really comparable (age/sex proportion, living conditions, COVID diagnosis), and otherwise to highlight the differences observed. Finally, he possible bias of the chosen representation is not discussed.

In the abstract and elsewhere after, it’s supported that “the use of personal protective equipment (PPE) are the only epidemiologic measures” while the facts exemplified may demonstrate opposite situations: please add also that PPE in our day-to-day lives are efficient when associated with frequent and proper hand washing.

There is too much citations from media and general press (e.g. BBC, YouTube…) in the R. Kellermayer’s article that could be replaced with scientific short communications or opinion views. A lot of references citations of key publications are lacking concerning the current epidemic of COVID-19 both in introduction and in discussion part (for example in the introduction: Lescure FX (https://doi.org/10.1016/S1473-3099(20)30237-1) in parallel to ref. [3] and He X https://www.nature.com/articles/s41591-020-0869-5 in parallel to ref. [4, 5]).

One point that particularly deserves to be developed in this article is the explanation of the interest especially in the BCG vaccination (even extension of use, see thereafter) in the context of COVID-19 epidemic: please explain the scientific rational.

Some evaluations in children of a BCG vaccine cross-protection against several types of microbial or fungal infection were published yet (doi:10.1016/j.cmi.2019.04.020 ; doi:10.3389/fimmu.2019.02806) and these studies are not mentioned or discussed. Likewise, there is no reference to the phase III clinical trials which have recently started (Holland (NCT04328441); Australia (NCT04327206)) to answer the question of the prophylactic effect of BCG vaccine on COVID-19, dedicated to health care, aiming at reducing the symptomatology and considered as an emergency measure and not a lasting solution.

In addition, many applications for prophylactic purposes currently take advantage of the anti-inflammatory response induced by this BCG vaccination, e.g. for type 1 diabetes, bladder cancer, or autoimmune diseases. This rational for use should also be discussed.

Reviewer #2: The authors explored the association between daily rates of COVID-19 case fatality and the presence of universal BCG vaccination.

At the time of submission of their manuscript, there was one preprint reporting a correlation between universal BCG vaccination policy and reduced morbidity and mortality for COVID-19, which has not published yet after peer-review (at least, it is not listed in pubmed as for today, the 2nd of May).

The current analysis defined BCG vaccination data based on the same database as the preprint they refer to. For the death rate data, they used the one reported on April 3rd (instead of March 31st) and they did not exclude countries with less than one million inhabitants.

They analyzed the association between the death rate or the Death Per Case (or case fatality rate)/Days from onset (dpc/d) to account for testing and time bias, and BCG vaccination. They also compared Netherlands, Spain, United Kingdom, France, Belgium, Portugal, and Germany (they called Modern "Colonial Era" countries to colonize America and Africa) to others.

They showed that there is significantly less death/million when BCG was introduced before 1980, but it is not the case when Death Per Case/Days of endemic (dpc/d) is considered. Finally, they reported that dpc/d is significantly higher in Modern Era Colonizer countries compared to others.

MAJOR RECOMMANDATIONS

The choice of 1980 has a cut off for having BCG was defined arbitrary. The authors state that “BCG would have affected the population of a country 40 years old and above [i.e. the population with increased vulnerability towards the SARS-CoV-2 infection]”. In studies reporting a non specific effect of live attenuated vaccines (such as BCG, oral polio, measles or smallpox), like the one published by Christine Benn and Peter Aaby, the heterologous protection against all cause morbidity and mortality is long-lasting, but as compared to the transient immediate defense mediated by the innate effector response. It is not as long-lasting as adaptive immune memory, and in no way it lasts 40 years. Moreover, the non potential beneficial non specific effect of BCG might be modulated with time, and other vaccines for instance. Thus, the authors could focus their analysis on a younger population, rather than the entire population of each of the Top 68 countries in terms of covid-19 induced death, or covid-19 severity.

The year of introduction of BCG vaccination can be one variable to study. However, the authors are certainly aware of the heterogeneity of the vaccine coverage depending on the country. Whether BCG vaccination is mandatory or not can result in different BCG vaccine coverage. In addition, the vaccine schedules and strains of BCG with different immunogenicity can differ between countries for instance. This should ideally be taken into account because it can be confounding, or at least be discussed by the authors.

MINOR RECOMMANDATIONS

Please update the introduction and discussion sections with the very recent literature about:

-The presence of viral RNA found in plasma and urine, and the successfully isolation of infectious viruses from these compartments (as for the presence of infectious particles in feces, there are inherent technical difficulties to isolate virus from feces because of the high risk of contamination of cell cultures with other microrganisms than the virus of interest)

-the transmission through aerosols

-the pathogenesis of SARS-CoV-2 (and not SARS-CoV) in non human primates, including the comparison between n=2 old and n=2 young animals

-there are currently several preprints on observationnal studies and the analysis of the association between BCG vaccination and covid-19, with positive, negative or no correlation

-there are now more than one trial to evaluate whether BCG could reduce the susceptibility to SARS-CoV2 infection and severity of covid-19

Please include a paragraph on the hypothesis as to how BCG could protect, or not, or even worsen the susceptibility to SARS-CoV2 infection and severity of covid-19, with few words on the concept of non specific effect of vaccines and on the new paradigm of trained immunity.

The quality of the figures is too low.

6. PLOS authors have the option to publish the peer review history of their article (what does this mean?). If published, this will include your full peer review and any attached files.

Reviewer #1: No

Reviewer #2: Yes: Anne-Sophie Beignon

---

## [Author Response · Author response to Decision Letter 0]

26 Jul 2020

Response to the Reviewers

The time and expertise of the reviewers is greatly appreciated. We responded to the valuable recommendations point-by-point below. The corresponding changes in the manuscript are highlighted with Word ‘track changes’. The modifications have greatly improved the quality of our manuscript. In general, we would like to underscore that our work is intended to demonstrate that the currently available epidemiological data on COVID-19 can not be used to make any reliable clinical prediction in regards to prevention or treatment against SARS-CoV-2, especially if examined as death/million. We wished to deliver this message in the most scientific way possible in order to effectively reach the biomedical research community. Hence, the “word of caution” in our title. The title has also been modified to more precisely cover our findings and message, which pertains to BCG epidemiology, and does NOT intend to evaluate the biologic evidence for BCG vaccination as an acute (i.e. short lived) protectant against COVID-19. We feel that PLOS ONE is and outstanding means to communicate this message. We added Giedrius Trakimas to our author list and performed additional analyses to more strongly support our conclusions.

Reviewer #1: The article from Richard Kellermayer’s group presents a critical analysis of the recent publication by Miller et al 2020 ref. [15] and proposes an original approach to establish a representation of the daily rates of COVID-19 case fatality (i.e. Death Per Case /Days of the endemic [dpc/d]). However, only strictly identical groups (age/sex proportion, living conditions, COVID diagnosis) could be strictly compared and would better answer the question of the effect of the BCG vaccine on current acute epidemic infection; could you address this question by adding an analysis of equivalent groups (for ex. in urban conditions)? Furthermore, to support the article statements, it is necessary to discuss whether the people who have benefited from BCG vaccination or not are really comparable (age/sex proportion, living conditions, COVID diagnosis), and otherwise to highlight the differences observed. Finally, he possible bias of the chosen representation is not discussed.

Response: Our general response partly addresses this recommendation of the distinguished Reviewer. We absolutely agree that if reliable and sound epidemiologic data were available, stricter analyses incorporating as many variables as possible would be desired. In the meantime, we have now performed multiple regression modeling and incorporated additional variables such as the recommended ‘urban living’ as predictors into our models. Our main conclusion has not been significantly modified by these analyses. We do highlight the arbitrary selection bias not only in respect to BCG vaccination (as the essence of our critique manuscript), but in regards to our ‘historical colonization status’ variable.

In the abstract and elsewhere after, it’s supported that “the use of personal protective equipment (PPE) are the only epidemiologic measures” while the facts exemplified may demonstrate opposite situations: please add also that PPE in our day-to-day lives are efficient when associated with frequent and proper hand washing.

Response: We decided to omit the discussions about PPE in respect to COVID-19, since that is not the focus of our work.

There is too much citations from media and general press (e.g. BBC, YouTube…) in the R. Kellermayer’s article that could be replaced with scientific short communications or opinion views. A lot of references citations of key publications are lacking concerning the current epidemic of COVID-19 both in introduction and in discussion part (for example in the introduction: Lescure FX (https://doi.org/10.1016/S1473-3099(20)30237-1) in parallel to ref. [3] and He X https://www.nature.com/articles/s41591-020-0869-5 in parallel to ref. [4, 5]).

Response: Thank you for the recommendations, we have incorporated the references. In the meantime, we would like to emphasize that the focus of our work is the critique of worldwide epidemiology data, not a comprehensive review of COVID-19 pathogenesis. 

One point that particularly deserves to be developed in this article is the explanation of the interest especially in the BCG vaccination (even extension of use, see thereafter) in the context of COVID-19 epidemic: please explain the scientific rational.

Some evaluations in children of a BCG vaccine cross-protection against several types of microbial or fungal infection were published yet (doi:10.1016/j.cmi.2019.04.020 ; doi:10.3389/fimmu.2019.02806) and these studies are not mentioned or discussed. Likewise, there is no reference to the phase III clinical trials which have recently started (Holland (NCT04328441); Australia (NCT04327206)) to answer the question of the prophylactic effect of BCG vaccine on COVID-19, dedicated to health care, aiming at reducing the symptomatology and considered as an emergency measure and not a lasting solution.

In addition, many applications for prophylactic purposes currently take advantage of the anti-inflammatory response induced by this BCG vaccination, e.g. for type 1 diabetes, bladder cancer, or autoimmune diseases. This rational for use should also be discussed.

Response: We have included a brief summary of the plausible biologic basis for BCG vaccination protection, and referenced a comprehensive and freely accessible web page based review on the topic in our discussion. We also included into our introduction the number of clinical trials registered in clinicaltrials.gov in June. However, our focus is to emphasize the highly biased and interdependent nature of arbitrary predictors of COVID-19 epidemiology, especially in review of the unreliable nature of country based case and death reporting.

Reviewer #2: The authors explored the association between daily rates of COVID-19 case fatality and the presence of universal BCG vaccination.

At the time of submission of their manuscript, there was one preprint reporting a correlation between universal BCG vaccination policy and reduced morbidity and mortality for COVID-19, which has not published yet after peer-review (at least, it is not listed in pubmed as for today, the 2nd of May).

The current analysis defined BCG vaccination data based on the same database as the preprint they refer to. For the death rate data, they used the one reported on April 3rd (instead of March 31st) and they did not exclude countries with less than one million inhabitants.

They analyzed the association between the death rate or the Death Per Case (or case fatality rate)/Days from onset (dpc/d) to account for testing and time bias, and BCG vaccination. They also compared Netherlands, Spain, United Kingdom, France, Belgium, Portugal, and Germany (they called Modern "Colonial Era" countries to colonize America and Africa) to others.

They showed that there is significantly less death/million when BCG was introduced before 1980, but it is not the case when Death Per Case/Days of endemic (dpc/d) is considered. Finally, they reported that dpc/d is significantly higher in Modern Era Colonizer countries compared to others.

MAJOR RECOMMANDATIONS

The choice of 1980 has a cut off for having BCG was defined arbitrary. The authors state that “BCG would have affected the population of a country 40 years old and above [i.e. the population with increased vulnerability towards the SARS-CoV-2 infection]”. In studies reporting a non specific effect of live attenuated vaccines (such as BCG, oral polio, measles or smallpox), like the one published by Christine Benn and Peter Aaby, the heterologous protection against all cause morbidity and mortality is long-lasting, but as compared to the transient immediate defense mediated by the innate effector response. It is not as long-lasting as adaptive immune memory, and in no way it lasts 40 years. Moreover, the non potential beneficial non specific effect of BCG might be modulated with time, and other vaccines for instance. Thus, the authors could focus their analysis on a younger population, rather than the entire population of each of the Top 68 countries in terms of covid-19 induced death, or covid-19 severity.

Response: Thank you for the insightful observations and recommendations. We absolutely agree with this Reviewer that there is no biologic basis for newborn/baby age BCG vaccination to protect against COVID-19 infection and/or morbidity in the elderly. We have now performed analyses on an additional (May 15th) dataset with multiple regression modeling, including median age as a predictor. We have also added a recent publication as reference (37) that contradicts newborn BCG effects on COVID-19, even in young adults around the age of 40. Our main conclusion is the unreliable nature of worldwide COVID-19 epidemiologic data and that researchers should consequently avoid, or highly critically perform detailed analyses of that.

The year of introduction of BCG vaccination can be one variable to study. However, the authors are certainly aware of the heterogeneity of the vaccine coverage depending on the country. Whether BCG vaccination is mandatory or not can result in different BCG vaccine coverage. In addition, the vaccine schedules and strains of BCG with different immunogenicity can differ between countries for instance. This should ideally be taken into account because it can be confounding, or at least be discussed by the authors.

Response: We would like to repeatedly emphasize the main point of our work, which underscores the unreliable nature of COVID-19 worldwide data. We trust that the added analyses and more clear discussion of those along with a stronger message to the epidemiologist community makes it understandable that we decided not to perform the recommended studies on the heterogeneity of the vaccine coverage, depending on the country.

MINOR RECOMMANDATIONS

Please update the introduction and discussion sections with the very recent literature about:

-The presence of viral RNA found in plasma and urine, and the successfully isolation of infectious viruses from these compartments (as for the presence of infectious particles in feces, there are inherent technical difficulties to isolate virus from feces because of the high risk of contamination of cell cultures with other microrganisms than the virus of interest)

-the transmission through aerosols

-the pathogenesis of SARS-CoV-2 (and not SARS-CoV) in non human primates, including the comparison between n=2 old and n=2 young animals

-there are currently several preprints on observationnal studies and the analysis of the association between BCG vaccination and covid-19, with positive, negative or no correlation

-there are now more than one trial to evaluate whether BCG could reduce the susceptibility to SARS-CoV2 infection and severity of covid-19

Please include a paragraph on the hypothesis as to how BCG could protect, or not, or even worsen the susceptibility to SARS-CoV2 infection and severity of covid-19, with few words on the concept of non specific effect of vaccines and on the new paradigm of trained immunity.

The quality of the figures is too low.

Response: We repeatedly hope that the added analyses, additional references and more clear discussion of those along with a stronger message to the epidemiologist community is acceptable to this Reviewer without us following all the minor recommendations. We wished to be as concise as possible and not elaborate more on viral transmission and animal modeling in respect to SARS-CoV-2 pathogenesis. We did include a paragraph referencing a rather recent comprehensive review on the plausible connections between BCG vaccination, and off target viral infections, including COVID-19. We also excluded our figures during the major revision of our work.

---

## [Decision Letter · Decision Letter 1]

15 Sep 2020

PONE-D-20-10246R1

BCG epidemiology supports its protection against COVID-19? A word of caution

PLOS ONE

Dear Dr. Kellermayer,

Thank you for submitting your manuscript to PLOS ONE. After careful consideration, we feel that it has merit but does not fully meet PLOS ONE’s publication criteria as it currently stands. Therefore, we invite you to submit a revised version of the manuscript that addresses the points raised during the review process.

as suggested by the reviewer, it might be of good practice to indicate the last paper about BCG and respiratory infection he highlighted.

We look forward to receiving your revised manuscript.

Kind regards,

Pierre Roques, Ph.D.

Academic Editor

PLOS ONE

Additional Editor Comments (if provided):

sorry for the delay but the end of shutdown and return to normal life is sometime quite complicated now;

Reviewers' comments:

Reviewer's Responses to Questions

**Comments to the Author**

1. If the authors have adequately addressed your comments raised in a previous round of review and you feel that this manuscript is now acceptable for publication, you may indicate that here to bypass the “Comments to the Author” section, enter your conflict of interest statement in the “Confidential to Editor” section, and submit your "Accept" recommendation.

Reviewer #2: All comments have been addressed

2. Is the manuscript technically sound, and do the data support the conclusions?

Reviewer #2: Yes

3. Has the statistical analysis been performed appropriately and rigorously? 

Reviewer #2: I Don't Know

4. Have the authors made all data underlying the findings in their manuscript fully available?

Reviewer #2: Yes

5. Is the manuscript presented in an intelligible fashion and written in standard English?

Reviewer #2: Yes

6. Review Comments to the Author

Reviewer #2: The authors have addressed the comments from both reviewers.

The goal of their work has been better defined (including by rephrasing the title of the paper and their conclusions). They highlight and alert that global COVID-19 epidemiological data (as well as global BCG epidemiological data?) are not reliable enough to base hypothesis-driven research.

After direct correlation analysis between the year of the establishment of universal BCG vaccination and the mortality rates, expressed as death/million or death per case/days, and using 2 datasets (from the beginning of April and mid of May), they now also used multiple regression analyses. They have included more variables, such as the urban population

percentage, population density, and air passengers, in addition to historic colonization status and median age.

The conclusions from the ongoing clinical trials testing whether BCG could protect against COVID-19 are not yet known, however a recent paper demonstrates that BCG can protect the elderly against respiratory infections (Giamarellos-Bourboulis et al., Cell, 31 Aug 2020). The authors might want to cite this reference as a stronger evidence to test the hypothesis rather than epidemiological data.

7. PLOS authors have the option to publish the peer review history of their article (what does this mean?). If published, this will include your full peer review and any attached files.

Reviewer #2: No

---

## [Author Response · Author response to Decision Letter 1]

20 Sep 2020

Reviewer #2: The conclusions from the ongoing clinical trials testing whether BCG could protect against COVID-19 are not yet known, however a recent paper demonstrates that BCG can protect the elderly against respiratory infections (Giamarellos-Bourboulis et al., Cell, 31 Aug 2020). The authors might want to cite this reference as a stronger evidence to test the hypothesis rather than epidemiological data.

Response: We agree, and have added an additional paragraph to the discussion, which critically reviews the recommended publication from our manuscript’s point of view. We also made minor modifications to the paper in order to more clearly emphasize that our work only pertains to the epidemiology of pediatric age delivered, universal BCG vaccination, and whether that may be protective against COVID-19 deaths in the adult/elderly population, who received the vaccine when they were very young (i.e. decades before a possible infection with COVID-19).

---

## [Editor Report · Decision Letter 2]

23 Sep 2020

BCG epidemiology supports its protection against COVID-19? A word of caution

PONE-D-20-10246R2

Dear Dr. Kellermayer,

We’re pleased to inform you that your manuscript has been judged scientifically suitable for publication and will be formally accepted for publication once it meets all outstanding technical requirements.

Kind regards,

Pierre Roques, Ph.D.

Academic Editor

PLOS ONE

Additional Editor Comments (optional):

Thanks to have take in acount all the referee comments. I congratulate you for this nice paper.
---

## [Editor Report · Acceptance letter]

25 Sep 2020

PONE-D-20-10246R2 

BCG epidemiology supports its protection against COVID-19? A word of caution 

Dear Dr. Kellermayer:

I'm pleased to inform you that your manuscript has been deemed suitable for publication in PLOS ONE. Congratulations! Your manuscript is now with our production department. 

Kind regards, 

on behalf of

Dr. Pierre Roques 

Academic Editor

PLOS ONE